# SpecEdge: Scalable Edge-Assisted Serving Framework for Interactive LLMs

**Jinwoo Park**
KAIST
jinwoo520528@kaist.ac.kr

**Seunggeun Cho**
KAIST
sgn.cho@kaist.ac.kr

**Dongsu Han**
KAIST
dhan.ee@kaist.ac.kr

## Abstract

Large language models (LLMs) power many modern applications, but serving them at scale remains costly and resource-intensive. Current server-centric systems overlook consumer-grade GPUs at the edge. We introduce SpecEdge, an edge-assisted inference framework that splits LLM workloads between edge and server GPUs using a speculative decoding scheme, exchanging only token outputs over the network. SpecEdge employs proactive edge drafting to overlap edge token creation with server verification and pipeline-aware scheduling that interleaves multiple user requests to increase server-side throughput. Experiments show SpecEdge enhances overall cost efficiency by **1.91×** through achieving **2.22×** server throughput, and reduces inter token latency by **11.24%** compared to a server-only baseline, introducing a scalable, cost-effective paradigm for LLM serving. The code is available at `https://github.com/kaist-ina/specedge`

## 1 Introduction

Large language models (LLMs) have become integral to modern applications such as conversational AI, code generation, and real-time content creation [Dubey et al., 2024, Lozhkov et al., 2024, Achiam et al., 2023, Touvron et al., 2023, Jiang et al., 2023, Brown et al., 2020]. However, scaling LLM deployments to meet growing demand remains challenging when balancing operational costs against latency requirements.

A compelling opportunity exists to dramatically reduce LLM serving costs by leveraging consumer-grade GPUs at the network edge. The GeForce RTX 4090 [Nvidia, 2025a] delivers up to 330.3 TFLOPS for FP16 tensor operations with FP16 accumulate [Nvidia, 2024a], exceeding the 312 TFLOPS of the data-center-class A100 [Nvidia, 2025b], at 14.43x lower cost [GCP, 2025, Vas, 2025]. With the widespread availability of these powerful edge devices [Valve, 2024, Nvidia, 2024b], an edge-assisted inference approach that offloads computation to these cost-effective resources could fundamentally transform the economics of LLM deployment.

Despite this opportunity, existing inference architectures fail to leverage these edge resources effectively. Current parallelization techniques [Shoeybi et al., 2019, Rasley et al., 2020, Aminabadi et al., 2022] that split computation within data centers break down over public internet conditions, where high latency and limited bandwidth make frequent communication of intermediate results impractical. Mainstream approaches like tensor and pipeline parallelism rely on high-bandwidth, low-latency interconnects such as NVLink or InfiniBand [Nvidia, 2025c,d]. In wide-area networks (WANs), transferring intermediate model states between edge and server GPUs quickly becomes prohibitive, preventing meaningful collaboration between these heterogeneous resources.

In this paper, we present SpecEdge, the first practical edge-assisted inference framework that fundamentally reduces LLM serving costs by splitting computation between consumer-grade edge GPUs and cloud servers. The core innovation of SpecEdge is its ability to effectively coordinate edge and server resources over wide-area networks—a capability previously unattainable with traditional

parallelization techniques. To achieve this, SpecEdge adopts a speculative decoding paradigm that divides LLM inference into *edge drafting* and *server verification*, exchanging only token outputs rather than full model states. This approach dramatically reduces bandwidth requirements while minimizing communication rounds.

To make this edge-assisted paradigm practical in real-world deployment scenarios, SpecEdge implements two key enabling techniques. Our *Proactive Edge Drafting* allows edge GPUs to continue generating tokens while awaiting server verification, effectively masking network and verification latency with local computation. Complementing this, our *Server-side Pipeline-aware Scheduling* orchestrates verification requests from multiple users through intelligent batching to maintain high GPU utilization. Together, these techniques ensure that the inherent cost advantages of edge-assisted inference are not undermined by network constraints or inefficient resource utilization.

Our evaluation validates the effectiveness of this edge-assisted approach, demonstrating that SpecEdge achieves **1.91×** better cost efficiency while increasing server-side throughput by **2.22×** and reducing inter token latency by **11.24%**. These improvements persist even under challenging wide-area network conditions, outperforming server-only baselines with zero network delays. By effectively harnessing widely available edge GPUs, SpecEdge establishes a new paradigm for scalable and cost-effective LLM serving that addresses the growing computational demands of generative AI applications.

## 2 Background and Related Work

**LLM serving systems.** Modern LLM frameworks address latency, throughput, and resource efficiency challenges through various optimizations. DeepSpeed-Inference [Rasley et al., 2020] and TensorRT-LLM [Nvidia, 2024c] leverage low-level GPU optimizations, model parallelism, and quantization, while vLLM [Kwon et al., 2023] introduces PagedAttention for efficient memory management. Modern parallelization strategies [Shoeybi et al., 2019, Aminabadi et al., 2022] enhance the performance with multiple GPUs. However, these approaches depend on data-center GPUs connected via specialized interconnects (InfiniBand, NVLink) with throughput exceeding hundreds of GB/s [Nvidia, 2025c]—speeds unattainable over wide area networks (WANs).

**Speculative decoding.** Speculative decoding [Leviathan et al., 2023, Chen et al., 2023] reduces latency by having a smaller auxiliary model generate multiple candidate tokens for parallel verification by the main model. The process involves three phases: drafting candidates, verification, and reconciliation for generated tokens. Recent advances have focused on more efficient drafting methods, either using lighter auxiliary models or the target model with reduced parameters [Bhendawade et al., 2024, Cai et al., 2024, Li et al., 2024, Stewart et al., 2024, Zhang et al., 2023].

**Tree-based speculative decoding.** Standard speculative decoding suffers from exponentially declining acceptance rates as sequence length increases [Leviathan et al., 2023, Chen et al., 2023]. Tree-based approaches [Miao et al., 2024, Chen et al., 2024, Svirschevski et al., 2024, Cai et al., 2024, Sun et al., 2024] address this by exploring multiple paths simultaneously. Sequoia [Chen et al., 2024] and SpecExec [Svirschevski et al., 2024] further optimize by pruning unpromising branches.

**Distributed LLM serving.** Split-inference approaches like Petals [Borzunov et al., 2024] and Helix [Mei et al., 2024] distribute model layers and pipeline requests across multiple devices, improving throughput over memory offloading methods [Ren et al., 2021, Pudipeddi et al., 2020], but introduce network delays that increase latency compared to data center solutions. Several works [Timor et al., 2025, Liu et al., 2025, McDanel, 2024] utilize multiple GPU devices within a server node to overlap draft and verification tasks of speculative decoding for faster inference, yet with an exchange of higher cost as they require additional server devices per query.

**On-device LLM inference.** Frameworks like MLC LLM [MLC team, 2023-2024] and Web LLM enable fully on-device inference with smaller or quantized models. While recent approaches [Svirschevski et al., 2024, Song et al., 2024, Xue et al., 2024] showcase the potential of leveraging user devices, they compromise output quality and latency. In contrast, our approach retains the high-quality verification stage on the server—achieving a hybrid solution that outperforms purely centralized or fully local inference.

# 3   Problem and Motivation

Serving large language models (LLMs) presents significant computational challenges due to their resource-intensive nature. While data centers rely on expensive H100 and A100 GPUs, abundant computing resources exist at the edge in the form of consumer-grade GPUs. As Figure 1 demonstrates, edge devices like the RTX 4090 and RTX 3090 generate tokens at approximately **30-50×** lower cost than server-class GPUs when running small but capable language models like Qwen2-0.5B. Despite this cost advantage and widespread availability, current LLM serving architectures fail to incorporate these edge resources, creating a substantial missed opportunity for distributed inference that could reduce costs while maintaining high-quality LLM service.

**Conventional split computing.** One approach to utilizing edge GPUs is split computation across edge and server resources. Split computing has been extensively studied in machine learning literature [Kang et al., 2017, Zhou et al., 2019, Wang et al., 2020], predating transformer architectures. Traditionally, this approach divides neural network layers between server and edge, reducing server computational load or minimizing network communication by transmitting intermediate tensors rather than raw input data. However, this method is poorly suited for LLM serving due to the unique characteristics of transformers [Vaswani, 2017]:

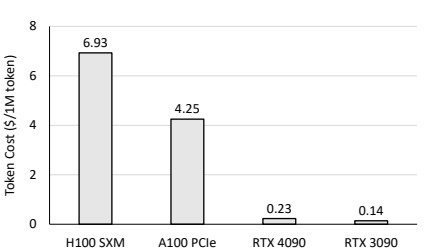

Figure 1: Token generation cost comparison with Qwen2-0.5B model.

**Limitation #1: Excessive latency.** While layer-wise splitting has been used in distributed peer-to-peer systems [Borzunov et al., 2024, Mei et al., 2024] to enable serving large models otherwise infeasible on consumer GPUs, these approaches prioritize feasibility over performance. This is because, unlike earlier ML applications [He et al., 2016, Redmon, 2016, Diwan et al., 2023] that typically involve only a few communication rounds, LLM applications require communication for every generated token due to their autoregressive nature. In our distributed network setting (§5.3), layer-wise splitting increases latency by 2.35x compared to the server-only solution.

**Limitation #2: Ineffectiveness in reducing I/O.** LLM inference is inherently constrained by memory I/O [Pope et al., 2023], as generating a single token requires accessing the entire model's parameters. Transformer architectures, with operations like general matrix-vector multiplication (GEMV), have low arithmetic intensity relative to their memory I/O demands. While layer-wise split computing reduces computational load by offloading layers to the edge, it fails to alleviate the I/O-bound nature of inference. A more effective approach is batch-verifying multiple candidate tokens, which significantly reduces the I/O overhead per token and increases GPU throughput.

# 4   SpecEdge Design

Our goal is to combine consumer-grade edge GPUs with server resources for cost-efficient LLM serving without compromising quality or latency.

## 4.1   Disaggregated LLM Decoding

To overcome conventional split computing limitations, we offload token drafting to edge GPUs while keeping verification on servers. Unlike traditional LLM serving, where servers handle the entire process, SpecEdge disaggregates speculative decoding into two distinct phases: edge-based *drafting* and server-based *verification*.

In our novel design, edge GPUs generate candidate tokens and send them to the server, which verifies them in a single forward pass. The server returns both the verified tokens and one additional token to the edge device, which then updates its sequence, KV cache, and continues drafting. This cycle repeats until an end-of-sequence token is generated. Critically, this approach preserves the exact output distribution of the server model [Leviathan et al., 2023]—even when using different models for drafting and verification, the final output is guaranteed to be sampled from the same distribution as if generated by the server model alone.

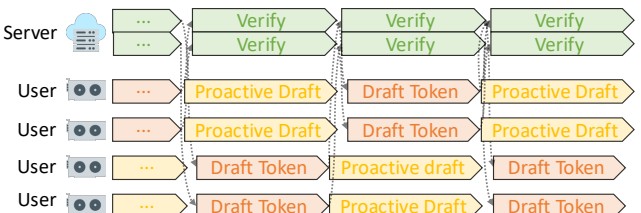

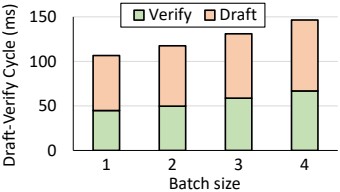

Figure 2: Abstract timeline of SpecEdge with *draft* (edge-side) and *verify* (server-side) inference concept.

Figure 3: A speculative decoding cycle composed of draft and verify stages.

As illustrated in Figure 2, this disaggregated approach enables the server to focus solely on verification while edge devices handle the autoregressive drafting process. By shifting much of the drafting time to inexpensive edge GPUs, we significantly reduce the overall serving cost while increasing server throughput.

Figure 3 quantifies the potential advantage of this approach by showing that draft stages account for the majority of the draft-verify cycle across different batch sizes. By offloading this dominant component to edge devices, SpecEdge can substantially reduce server time and improve overall system efficiency.

This disaggregated approach promises two fundamental advantages over conventional split computing:

- **Resolving excessive latency (Limitation #1):** Unlike layer-wise splitting that requires per-token communication rounds, SpecEdge dramatically reduces network interaction by a speculative decoding approach. This minimizes both the number of client-server round trips and the volume of data transferred, eliminating the excessive latency inherent in traditional split approaches.
- **Addressing I/O bottlenecks (Limitation #2):** By enabling batch verification of multiple tokens in a single server-side forward pass, SpecEdge amortizes the cost of model parameter access across multiple token verifications. This increases the arithmetic intensity of each operation, directly addressing the I/O-bound nature of LLM inference and substantially improving GPU throughput.

However, to fully realize these benefits, we must overcome two critical challenges that emerge when separating drafting and verification across different devices:

- **Potential latency increase:** In conventional speculative decoding, drafting and verification happen sequentially on the same device with minimal transition overhead. In our disaggregated setting, naïvely implementing this sequence would add network round-trip delays to each draft-verify cycle, potentially negating our latency advantages and deteriorating user experience.
- **Risk of server underutilization:** Without careful coordination, server GPUs would remain idle while waiting for edge devices to complete drafting. This inefficiency could severely limit throughput and undermine the cost benefits of our approach, particularly when serving multiple concurrent users with varying workload patterns.

To unlock the full potential of disaggregated speculative decoding, SpecEdge introduces two key innovations: Proactive edge drafting (§4.2), which masks network latency by continuously generating token candidates without waiting for verification results, and pipeline-aware scheduling (§4.3), which maximizes server GPU utilization by intelligently batching verification requests from multiple users. Together, these techniques enable SpecEdge to achieve both low latency and high cost efficiency.

## 4.2 Proactive Draft Generation at the Edge

Our key insight is to eliminate the idle time by continuing draft generation during server verification. The edge GPU performs two types of drafting: initial drafting to generate the first batch of candidate tokens, and proactive drafting that continues during server verification. When the edge GPU sends $n$ candidate tokens to the server, it immediately continues drafting additional tokens without waiting for verification results. If any token is rejected during verification, these proactively generated tokens are discarded. However, when all $n$ tokens are accepted and the server's bonus token matches the first proactively drafted token—a scenario we call "complete draft alignment"—these additional tokens can be immediately utilized, effectively hiding network and verification latency. This approach significantly reduces end-to-end latency by overlapping computation with communication, eliminating waiting periods between draft-verify cycles. Simultaneously, it improves server throughput by

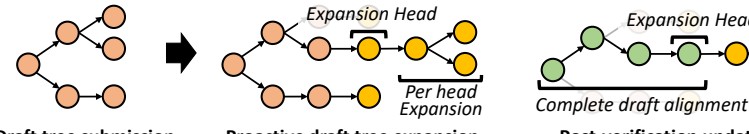

**Figure 4:** Illustrative example of a proactive draft tree expansion.

**Figure 5:** Post-verification update with complete draft alignment and subsequent draft submission.

ensuring a continuous flow of verification requests, keeping expensive server GPUs highly utilized rather than idle while waiting for edge devices to complete drafting.

**Initial drafting.** SpecEdge employs state-of-the-art tree-based drafting [Svirschevski et al., 2024]. However, SpecEdge is designed to be future-proof and not tied to any specific drafting technique. It simply requires a draft phase that produces candidate tokens and is compatible with various speculative decoding approaches, including tree-based methods [Chen et al., 2024, Miao et al., 2024], lossy/lossless methods, and earlier single-sequence schemes [Leviathan et al., 2023, Chen et al., 2023]. As the field evolves, any advances in speculative decoding can be seamlessly integrated.

**Proactive draft tree expansion.** Unlike conventional speculative decoding that stops after initial token generation, our approach continues drafting proactively to anticipate future server responses. The expected gain from this proactive expansion can be expressed as:

$$\mathbb{E}(\text{Gain}) = P_{\text{align}} \cdot P_{\text{match | align}} \cdot \left( \frac{T_{\text{draft}}}{H_{\text{expan}}} - 1 \right) \tag{1}$$

where $P_{\text{align}}$ represents the probability of alignment between verified and drafted sequences, $P_{\text{match | align}}$ is the probability that the server's extra token matches an expansion head given alignment, $T_{\text{draft}}$ is the total number of proactively drafted tokens, and $H_{\text{expan}}$ is the number of expansion heads (leaf nodes from which drafting continues).

This formulation reveals a fundamental trade-off: increasing $H_{\text{expan}}$ improves alignment probability ($P_{\text{align}}$) but decreases the token preservation ratio ($\frac{T_{\text{draft}}}{H_{\text{expan}}} - 1$), while fewer heads reduce alignment probability but significantly increase preservation when alignment succeeds. Naïvely applying the initial drafting strategy—treating every leaf node as an expansion head—yields negligible gains despite maximizing $P_{\text{align}}$. Our empirical results demonstrate a counterintuitive but optimal approach: after generating the initial draft tree, SpecEdge identifies the single path with the highest cumulative log probability and continues drafting exclusively from that node (Figure 4). This focused strategy maximizes expected gain by producing significantly higher returns when alignment occurs, despite the reduced alignment probability.

**Post-verification update.** When server verification completes, the edge GPU receives both the accepted tokens and one additional token generated during the server's forward pass. At this point, SpecEdge compares the server's output with its proactively drafted tokens. Also, the edge updates its draft model KV cache according to the accepted tokens.

If the server's accepted tokens and additional token perfectly match a path in the proactively drafted tree—a complete draft alignment—SpecEdge retains that branch and allows the edge to continue drafting from this advanced position (Figure 5). This approach eliminates the need to regenerate already-drafted tokens, generating a deeper draft for the next drafting round by efficiently reusing prior computations.

If complete alignment fails—either because the verified path diverges from the proactive tree or doesn't reach a leaf node—SpecEdge discards the proactive work and reverts to initial drafting, rebuilding the tree from the end of the verified sequence. While this scenario doesn't benefit from proactive drafting, the strategy still improves average performance by exploiting the successful alignments when they occur.

## 4.3 Server-side Pipeline-aware Scheduling

Unlike conventional speculative decoding, where servers perform both drafting and verification, SpecEdge dedicates server GPUs exclusively to verification. This separation allows the server to focus on batch-based verification rather than token-by-token generation, significantly improving

resource utilization. However, this disaggregated approach creates a new scheduling challenge: while the edge is busy drafting the next set of tokens, the server would idle if it only awaited verification tasks from that same request. This pattern creates "bubbles" of unutilized compute capacity whenever one part of the system is waiting on the other.

**Pipeline-aware verification scheduling.** SpecEdge eliminates computational inefficiencies by interleaving verification tasks across multiple requests processed on separate edge devices. The server continuously verifies completed draft batches from one set of requests while other requests simultaneously generate new drafts on their respective edge devices. This pipelined approach ensures immediate processing of incoming drafts, with verified requests promptly returning to their edge devices for additional drafting, thereby freeing server resources for the next verification batch. By aligning edge device count with server verification capacity, this orchestration effectively doubles server throughput compared to conventional server-only configurations of equivalent batch size, substantially enhancing both cost efficiency and GPU utilization.

For optimal pipeline efficiency, SpecEdge dynamically calibrates the relationship between server verification time and edge operations using real-time performance measurements. The system adjusts draft depth—the number of forward passes through the draft model—to satisfy the equation: server verification time ≈ edge drafting time + network round-trip time. This calibration ensures token batches arrive at the server precisely as it completes verifying previous batches, eliminating computational bubbles and minimizing end-to-end latency while maintaining maximum resource utilization across the distributed system.

**Processing heterogeneous requests.** Server-side verification must efficiently handle batches containing requests with varying sequence lengths—a common scenario when multiple users are at different stages in their generation process. SpecEdge addresses this challenge through two complementary techniques. First, it employs custom attention masking for each token sequence in the batch, ensuring the model attends only to valid tokens within that sequence while enabling parallel processing without cross-sequence interference. Second, it implements KV cache padding to match the longest sequence in the batch, avoiding the substantial computational cost of reconfiguring tensor shapes during inference despite minimal overhead for shorter sequences.

This dual approach allows SpecEdge to process diverse verification requests in unified batches, fully leveraging GPU parallelism while accommodating the asynchronous nature of edge-to-server communication. The result is maximized server throughput without sacrificing the responsiveness essential for interactive applications.

## 5 Evaluation

We evaluate SpecEdge in an edge-assisted server configuration against a server-only configuration across various LLMs and datasets. Our findings are summarized as follows:

- SpecEdge enhances cost efficiency by an average of **1.91×** compared to the server-only environment through increasing server throughput by **2.22×** on average.
- It reduces the inter token latency by an average of **11.24%**, even with a 14.07 ms round-trip time between the server and edge, outperforming the server-only configuration with no network delay.

**Implementation and Setup.** Our system's edge-assisted configuration utilizes a server-side NVIDIA A100 GPU connected to multiple edge-side NVIDIA RTX4090 GPUs over a wide-area network. The number of RTX4090 GPUs scales with the number of concurrent requests (batch size x 2). In our experiments, we measured an average round-trip time (RTT) of 14.07ms between the local edge node and our Google Cloud instance. We conducted evaluations across various models and datasets under diverse operating conditions. The code is available at `https://github.com/kaist-ina/specedge`

**Baseline and metrics.** Our primary baseline is a server-only configuration employing tree-based speculative decoding, supplemented by autoregressive decoding and a layer-split approach that offloads part of the LLM's layers to an edge device. SpecEdge can leverage either client-side or edge GPUs; in this evaluation, we assume it uses consumer-grade GPUs from edge cloud providers. Based on provider pricing [GCP, 2025, Vas, 2025], the server-side A100 40GB GPU costs $4.05 per hour, while running the 32B model requires an A100 80GB at $5.05 per hour. In comparison, SpecEdge adds $0.35 per hour for each RTX 4090 used. Our key metrics include cost efficiency

Table 1: Throughput and cost efficiency comparison between SpecEdge and server-only method.

| Target/Draft | Task | Gen. tokens per verify | | Server Throughput (tok/s) | | Cost Efficiency (1k toks/$) | |
|---|---|---|---|---|---|---|---|
| | | Server-only | **SpecEdge** | Server-only | **SpecEdge** | Server-only | **SpecEdge** |
| Qwen3 14B/1.7B | Multi-turn bench | $3.92\pm1.51$ | $3.98\pm1.57$ | 31.78 | 66.54 (**2.09x**) | 28.25 | 50.60 (**1.79x**) |
| | Translation | $3.95\pm1.47$ | $4.25\pm1.45$ | 32.24 | 65.25 (**2.02x**) | 28.66 | 49.47 (**1.73x**) |
| | Summarization | $3.73\pm1.60$ | $3.95\pm1.61$ | 29.70 | 67.53 (**2.27x**) | 26.40 | 51.22 (**1.94x**) |
| | QA | $3.42\pm1.57$ | $3.59\pm1.56$ | 27.30 | 62.04 (**2.27x**) | 24.26 | 47.09 (**1.94x**) |
| | Math. | $4.10\pm1.48$ | $4.25\pm1.49$ | 32.84 | 72.93 (**2.22x**) | 29.19 | 55.28 (**1.89x**) |
| | RAG | $3.73\pm1.53$ | $3.83\pm1.56$ | 29.89 | 64.04 (**2.14x**) | 26.57 | 48.78 (**1.84x**) |
| Qwen3 14B/0.6B | Multi-turn bench | $3.87\pm1.41$ | $4.41\pm2.25$ | 33.45 | 69.58 (**2.08x**) | 29.73 | 52.97 (**1.78x**) |
| | Translation | $3.79\pm1.48$ | $4.67\pm2.34$ | 32.88 | 69.00 (**2.10x**) | 29.22 | 52.23 (**1.79x**) |
| | Summarization | $3.68\pm1.49$ | $4.21\pm2.16$ | 31.17 | 68.60 (**2.20x**) | 27.71 | 52.16 (**1.88x**) |
| | QA | $3.33\pm1.41$ | $3.79\pm1.94$ | 28.89 | 61.90 (**2.14x**) | 25.68 | 46.98 (**1.83x**) |
| | Math | $3.90\pm1.57$ | $5.27\pm2.27$ | 33.53 | 83.88 (**2.50x**) | 29.80 | 63.56 (**2.13x**) |
| | RAG | $3.53\pm1.52$ | $4.29\pm2.16$ | 30.07 | 69.51 (**2.31x**) | 26.73 | 52.76 (**1.97x**) |
| Qwen3 32B/1.7B | Multi-turn bench | $4.22\pm1.93$ | $4.71\pm2.66$ | 24.96 | 56.47 (**2.26x**) | 17.80 | 35.38 (**1.99x**) |
| | Translation | $4.08\pm1.97$ | $5.24\pm2.84$ | 24.33 | 58.79 (**2.42x**) | 17.34 | 36.83 (**2.12x**) |
| | Summarization | $4.19\pm2.01$ | $4.52\pm2.68$ | 24.33 | 54.07 (**2.42x**) | 17.65 | 33.90 (**2.12x**) |
| | QA | $3.62\pm1.95$ | $3.93\pm2.49$ | 21.59 | 46.14 (**2.14x**) | 15.39 | 28.99 (**1.88x**) |
| | Math. | $4.60\pm1.93$ | $5.40\pm2.78$ | 27.52 | 64.01 (**2.33x**) | 19.62 | 40.18 (**2.05x**) |
| | RAG | $3.89\pm2.05$ | $4.19\pm2.69$ | 22.67 | 49.46 (**2.18x**) | 16.16 | 31.05 (**1.92x**) |

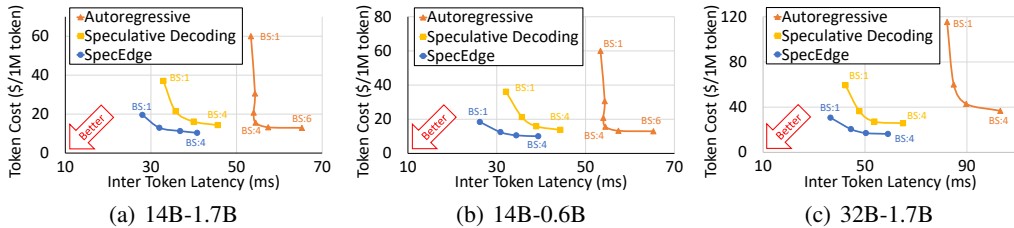

(a) 14B-1.7B  (b) 14B-0.6B  (c) 32B-1.7B

Figure 6: Per token cost and inter token latency comparison between server-only baselines and SpecEdge with varying batch size (BS) and model pairs.

(generated tokens per dollar), server throughput (generated tokens per unit time) and inter token latency (user-perceived output latency). All configurations produce identical output distributions as they use the same underlying models.

**Models and data sets.** We use four different LLMs: Qwen3-32B/14B [Team, 2025], Vicuna-33B [Chiang et al., 2023] and Llama2-13B-chat-hf [Touvron et al., 2023]. Unless specifically noted, all models are configured with a temperature setting of 0.7. For the draft models, we use five different models: Qwen3-1.7B/0.6B [Team, 2025], Sheared Llama-1.3B [Xia et al., 2023], Tiny Llama-1.1B [Zhang et al., 2024], and JackFram-160M [Miao et al., 2024]. Finally, we use SpecBench [Xia et al., 2024], C4 (en) [Raffel et al., 2020], OpenAssistant conversations datasets [Köpf et al., 2024], WikiText-2 [Merity et al., 2016], and MTBench [Zheng et al., 2023].

## 5.1 End-to-end Performance and Cost-efficiency

Table 1 presents a comparison of server-side throughput and cost efficiency between SpecEdge and a server-only speculative decoding baseline on six SpecBench tasks. We use a batch size of 1, which shows the lowest inter token latency for both SpecEdge and server-only baseline, appropriate for the latency-sensitive interactive LLM serving. Throughout the end-to-end evaluation, SpecEdge achieves **1.91×** better cost efficiency on average through **2.22×** throughput gain compared to the server-only setup. Despite the slight cost increase of employing consumer-grade edge GPUs,

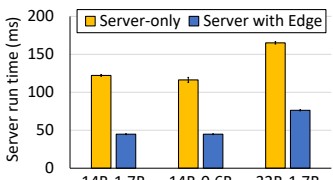

Figure 7: Server run time between server-only and server with edge drafting.

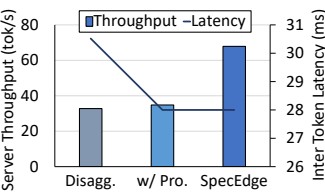

Figure 8: Performance comparison with SpecEdge components ablation.

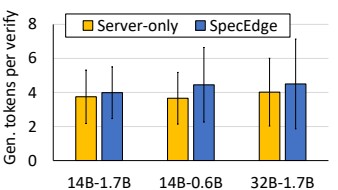

Figure 9: Generated tokens per verification between server-only and SpecEdge.

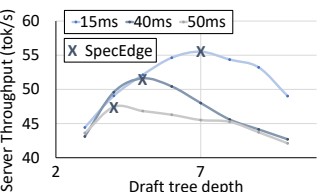

Figure 10: Server throughput according to draft depth under various network latencies.

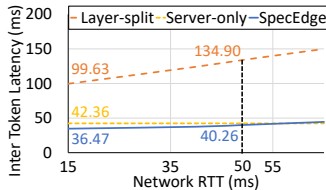

Figure 11: Inter token latency according to network round-trip time.

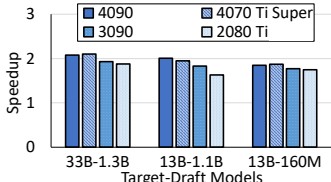

Figure 12: SpecEdge speedup gain with various edge devices upon autoregressive decoding.

SpecEdge's significant increase in server-side throughput ultimately leads to greater cost efficiency. This throughput gain is driven by server-side pipeline-aware scheduling (§4.3), which interleaves multiple requests that effectively increase server utilization, and by proactive edge drafting (§5.2), which increases the average number of generated tokens per verification. We also report the mean and standard deviation of tokens produced per verification cycle.

Beyond throughput improvements, Figure 6 demonstrates that SpecEdge also reduces inter token latency by an average of 11.24% compared to server-only speculative decoding across various batch sizes. This dual benefit—higher throughput with lower latency—is particularly valuable for interactive LLM applications where both resource efficiency and user experience are critical.

We extend our evaluation to other models, using Vicuna-33B and Llama2-13B-Chat as target LLMs across diverse datasets including C4, OAsst, WikiText-2, and MTBench. The results show that proactive edge drafting SpecEdge consistently improves performance, with greater benefits observed when the drafting models generate deeper drafts with better alignment toward the target LLMs. This alignment quality directly correlates with improvements in server throughput and reduced inter token latency. The complete results for all combinations of models and sets are provided in the Appendix B.1.

## 5.2 Component-wise Benefit

As illustrated in Figure 7, the server-only baseline devotes resources to both drafting and verification, causing prolonged server occupancy with each draft-verify cycle. In contrast, SpecEdge dedicates the server to verification alone while offloading drafting to more cost-effective edge devices, reducing server runtime by approximately 40-50% in all model configurations.

However, simple disaggregation alone can increase latency and leave the server underutilized (§4.1). Figure 8 compares three progressive implementations with a 14B/1.7B model pair: basic disaggregation, disaggregation with proactive edge drafting (§4.2), and complete SpecEdge with pipeline-aware scheduling (§4.3). Basic disaggregation achieves only 32.76 tokens/s throughput with higher latency, while adding proactive drafting reduces inter token latency to 28 ms. The complete SpecEdge with pipeline-aware scheduling dramatically increases server throughput to 67.89 tok/s (2.07× improvement) while maintaining latency, demonstrating the complementary benefits of each component.

**Generated tokens per verification.** Figure 9 compares the number of generated tokens per verification cycle between server-only speculative decoding and SpecEdge across three target–draft model pairs. On average, SpecEdge achieves 13.21% higher tokens per verification round. For the 32B–1.7B configuration, SpecEdge produces 4.5 tokens per verification compared to 4.02 with the server-only approach, while the 14B–0.6B pair sees similar gains (4.45 vs. 3.66). This efficiency gain stems from

proactive draft tree expansion, where each complete draft alignment allows for deeper draft trees in subsequent rounds, significantly enhancing verification efficiency and overall system performance.

**Pipeline-aware draft depth adjustment.** We demonstrate that dynamically adjusting the number of forward passes for edge drafting, as outlined in (§4.3), aligns the drafting phase with both server verification and network round-trip times, achieving optimal throughput in practice. Figure 10 shows how server-side throughput varies with draft tree depth under different RTTs using a 32B/1.7B model pair. On average, verification takes 94.2 ms, while each draft model forward pass needs about 11 ms. When RTT is 15 ms, SpecEdge sets the draft depth to seven; at 40 ms RTT, it sets the depth to five; and at 50 ms RTT, it decreases further to four. These results show that SpecEdge adapts draft depth for peak throughput across a range of network conditions.

## 5.3 System Sensitivity Analysis

**Network RTT sensitivity.** We evaluate the average inter token latency of SpecEdge against layer-split and server-only configurations across varying network round-trip times (RTTs), using Qwen32B on SpecBench. Layer-split configuration runs autoregressive decoding, where one-quarter of model layers run on an edge RTX4090, with the remainder on a server-side A100. The server-only configuration, unaffected by network RTT, runs tree-based speculative decoding entirely on an A100. Figure 11 shows that SpecEdge provides lower inter token latency than the server-only baseline below 50 ms RTT, with a 13.90% gain at 15 ms RTT (36.47 ms vs. 42.36 ms). Even at 65 ms, SpecEdge 's latency rises by only 22.00% over its 15 ms RTT performance (to 44.47 ms), remaining competitive. By contrast, layer-split is much slower: at 15 ms RTT, it is 2.73× slower than SpecEdge (99.63 ms vs. 36.47 ms), increasing to 3.35× slower performance at 50 ms RTT (134.90 ms vs. 40.26 ms). This resilience stems from SpecEdge 's less frequent communication rounds and proactive edge drafting, which offset network latency more effectively than layer-split approaches.

**Performance with varying edge devices.** Figure 12 presents the speedup achieved by SpecEdge when the server is assisted by different edge devices (RTX 4090, 4070 Ti Super, 3090, and 2080 Ti) across three target-draft model combinations. The speedup is measured relative to default autoregressive decoding using only the A100 server GPU. The consistent speedup across all model combinations confirms the architecture's robustness to different hardware configurations. As expected, more powerful edge GPUs like the RTX 4090 deliver greater speedups, while even more affordable options like the RTX 2080 Ti still provide significant acceleration. This demonstrates that SpecEdge's approach remains effective across a spectrum of edge hardware capabilities, allowing deployment flexibility. Additional results with lighter GPUs (3060 Ti, and 2080 Ti) are available in Appendix B.2.

**Performance with alternative drafting approaches.** We also evaluated SpecEdge with a non-tree speculative decoding approach [Leviathan et al., 2023] to demonstrate versatility beyond tree-structured methods. Using this alternative architecture, SpecEdge achieved up to 1.96× higher server throughput (Llama2 13B/TL 1.1B pairing) and 1.67× better cost efficiency compared to server-only deployments. Performance gains remained consistent across different model combinations, with even the smallest draft model (JF 160M) delivering a 1.52× throughput improvement while preserving end-user speedup. Complete results are available in Appendix B.3.

**Performance with batch drafting method.** We explore an alternative drafting configuration where a single edge GPU serves concurrent requests through batching. This approach enables operators to reduce the number of edge GPUs. Experiment with various batch sizes revealed a trade-off between better cost efficiency and increased latency. This configuration could be advantageous in budget-constrained deployments where latency tolerance is higher. Full results across model combinations are available in Appendix B.4.

**Performance under reasoning mode.** Modern LLMs provide reasoning mode for enhancing output quality, where reasoning tokens might influence speculative decoding efficiency [Wei et al., 2023, Team, 2025]. To explore SpecEdge performance under reasoning mode, we measured accepted tokens, server throughput, and cost efficiency with and without reasoning enabled. Results in Appendix B.5 show consistent improvements across all three metrics when reasoning is active, implying that our system inherently benefits from the redundancy in reasoning processes, which enhances speculative decoding performance.

**Cost analysis with various GPU Providers.** To ensure SpecEdge's cost-efficiency findings generalize beyond a single provider, we validated SpecEdge's performance across diverse cloud environments.

We compared results using GPUs from multiple GPU providers (Vast.ai [Vas, 2025], Runpod [run, 2025], and TensorDock [Ten, 2025]) and Cloud Service Providers (Google Cloud Platform [GCP, 2025], Amazon Web Services [AWS, 2025], and Microsoft Azure [Azu, 2025]), accounting for the pricing variations. Across all tested configurations, SpecEdge consistently delivered cost efficiency improvements, confirming that our architectural benefits persist regardless of the specific cloud infrastructure. Detailed cross-provider comparisons are presented in Appendix C.

**Detailed Case Study.** Complementing our quantitative evaluation, Appendix D offers a visualization of SpecEdge in action. Using Llama models responding to a query about Dyson Spheres, we trace the complete token generation lifecycle—from initial drafting through verification and subsequent accelerated generation. The case study specifically highlights two key operational advantages: (1) how edge GPUs remain productive during server verification phases through proactive expansion, and (2) how successful draft alignments lead to deeper draft candidates in subsequent rounds.

# 6  Conclusion

We have presented SpecEdge, an edge-assisted LLM inference framework that leverages user-side consumer-grade GPUs for drafting candidate tokens, while the server focuses on final verification. By transmitting only finalized outputs, SpecEdge efficiently operates under typical wide-area network conditions. Proactive edge drafting on the user side maximizes the utilization of edge GPUs and reduces end-to-end latency, while pipeline-aware verification scheduling at the server ensures high throughput by efficiently aggregating and processing verification requests. Our experimental results demonstrate that SpecEdge significantly reduces operational costs by **1.91×** through delivering **2.22×** higher server throughput, and achieves a modest reduction in latency compared to server-only baselines. Overall, SpecEdge unlocks the untapped potential of powerful consumer GPUs at the edge, offering a scalable and cost-effective approach for future LLM serving deployments.

## Broader Impact

This work redefines the division of labor in large language model (LLM) serving by integrating edge-side draft token generation with server-side verification, moving beyond the conventional centralized paradigm. This paradigm shift not only boosts server throughput without requiring additional data center infrastructure but also enables a novel business model for LLM services. By harnessing edge GPUs—whether through user-owned devices or edge cloud providers—our approach reduces reliance on expensive centralized servers, allowing service providers to deliver scalable, high-performance inference at a fraction of the cost. This decentralized architecture empowers businesses to adapt their infrastructure to edge resources, unlocking more flexible and cost-effective deployment strategies. Furthermore, SpecEdge alleviates the cost constraints associated with drafting, opening new avenues for speculative decoding research. By enabling the development of richer and more precise draft token generation methods, it advances the performance and capabilities of LLM services.

## Limitation and Future Work

SpecEdge is designed with flexibility in mind, supporting scenarios where users may contribute their own GPUs for the edge drafting phase. Our cost-efficiency analysis focuses on deployments using consumer-grade GPUs rented from edge cloud providers, but user-owned GPUs could further enhance cost savings and scalability. Extending SpecEdge to fully leverage user-operated hardware opens up exciting opportunities for decentralized and community-driven inference. At the same time, such scenarios raise new challenges in areas such as fault tolerance and security when untrusted devices participate in computation. While SpecEdge already supports a distributed multi-user environment, exploring these broader system and security aspects is an important direction for future work.

## Acknowledgment

We thank the anonymous reviewers for providing helpful feedback and suggestions to improve our work. This work was supported by Institute of Information & Communications Technology Planning & Evaluation (IITP) of the Korea government (MSIT) (No. RS-2024-00398157).

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

# A  Additional Details on Evaluation Setup

For both SpecEdge and the server-only speculative decoding baseline, we adopt a tree construction algorithm from recent tree-based speculative decoding [Chen et al., 2024, Svirschevski et al., 2024]. Given a specified tree size, each forward pass of the draft model generates multiple parallel candidate tokens, which are then pruned based on cumulative log probabilities so that the total number of tokens remains within the tree budget. The total number of forward passes (the draft tree depth) is set as a hyperparameter. In our main experiments, we use the draft tree size to 32 for each request. For the server-only baseline, we find and select the optimal draft tree depth through exhaustive search, while SpecEdge determines its draft depth based on verification and network latencies as described in Section 4.3.

Throughout our evaluation, we use off-the-shelf target LLMs and draft models without any additional fine-tuning. For the main experiments, we employ Qwen3 32B/14B as the target LLM and Qwen3 1.7B/0.6B as the draft models, using the SpecBench dataset. To demonstrate broader applicability, the appendix includes experiments with Vicuna 33B and Llama2-13B-Chat-hf as target LLMs, alongside Sheared Llama 1.3B, Tiny Llama 1.1B, and JackFram 160M as draft models, evaluated on C4, OAsst, WikiText-2, and MTBench. For each query, we generate up to 256 output tokens. Since the prefill stage in SpecEdge can involve parallel processing on both server and edge, the prefill latency is determined by $\max(\text{server\_time}, \text{edge\_time})$ which makes metrics such as time to first token comparable to the baseline; therefore, our reported metrics focus on the output tokens after prefill. Unless otherwise specified, SpecBench (spanning six different tasks) is used throughout the experiments.

# B  Comprehensive Performance Analysis

## B.1  Performance with Diverse Models and Datasets

In Table 2, we showcase SpecEdge 's adaptability across different target–draft model pairs and datasets, using a batch size of 1. We measure server throughput and cost efficiency gains, as well as the speedup achieved in inter token latency. Table 3 compares these speedups for both SpecEdge and a server-only speculative decoding baseline against an autoregressive decoding reference.

Our evaluation demonstrates the effectiveness of SpecEdge across several models including Qwen, Vicuna, and Llama with parameters up to 33B. While exploration with even larger models is planned as future work, we anticipate that no fundamental design changes will be needed for scaling. The

Table 2: Throughput and cost efficiency comparison between SpecEdge and server-only method.

| Target/Draft | Dataset | Gen. tokens per verify | | Server Throughput (tok/s) | | Cost Efficiency (1k toks/$) | |
|---|---|---|---|---|---|---|---|
| | | Server-only | **SpecEdge** | Server-only | **SpecEdge** | Server-only | **SpecEdge** |
| Vicuna 33B /SL 1.3B | C4 | 3.47±1.41 | 4.41±2.03 | 28.12 | 49.96 (**1.78x**) | 24.99 | 38.23 (**1.53x**) |
| | OAsst | 3.55±1.37 | 4.35±2.06 | 28.72 | 49.94 (**1.74x**) | 25.53 | 37.89 (**1.48x**) |
| | WikiText-2 | 3.42±1.45 | 4.19±1.98 | 27.26 | 47.94 (**1.76x**) | 24.23 | 36.42 (**1.50x**) |
| | MTBench | 3.62±1.72 | 4.57±2.13 | 29.62 | 52.70 (**1.78x**) | 26.33 | 40.21 (**1.53x**) |
| Llama2 13B /TL 1.1B | C4 | 3.40±1.35 | 3.48±1.34 | 36.80 | 70.76 (**1.92x**) | 32.72 | 53.72 (**1.64x**) |
| | OAsst | 3.57±1.40 | 3.67±1.39 | 38.55 | 69.49 (**1.80x**) | 34.27 | 52.75 (**1.54x**) |
| | WikiText-2 | 3.52±1.37 | 3.61±1.40 | 37.69 | 70.21 (**1.86x**) | 33.50 | 53.30 (**1.59x**) |
| | MTBench | 3.67±1.43 | 3.75±1.46 | 40.96 | 76.69 (**1.87x**) | 36.41 | 58.21 (**1.60x**) |
| Llama2 13B /JF 160M | C4 | 3.03±1.34 | 3.13±1.37 | 42.72 | 66.20 (**1.55x**) | 37.98 | 50.36 (**1.33x**) |
| | OAsst | 2.79±1.37 | 2.85±1.37 | 39.19 | 56.52 (**1.44x**) | 34.84 | 42.95 (**1.23x**) |
| | WikiText-2 | 2.72±1.37 | 2.72±1.39 | 37.87 | 55.39 (**1.46x**) | 33.66 | 42.08 (**1.25x**) |
| | MTBench | 2.80±1.39 | 2.80±1.43 | 40.80 | 60.42 (**1.48x**) | 36.42 | 45.92 (**1.26x**) |

Table 3: Speedup of server-only, SpecEdge compared to autoregressive.

| Target/Draft Model | Dataset | ITL (ms) | | Speedup | |
| --- | --- | --- | --- | --- | --- |
| | | Server-only | **SpecEdge** | Server-only | **SpecEdge** |
| Vicuna 33B/SL 1.3B | C4 | 33.229 | 30.612 | 1.86x | 2.02x |
| | OAsst | 34.496 | 30.460 | 1.76x | 2.00x |
| | WikiText-2 | 34.77 | 30.631 | 1.69x | 1.92x |
| | MTBench | 33.948 | 31.168 | 1.82x | 1.98x |
| Llama2 13B/TL 1.1B | C4 | 23.687 | 21.851 | 1.63x | 1.76x |
| | OAsst | 24.661 | 22.377 | 1.54x | 1.70x |
| | WikiText-2 | 23.883 | 21.338 | 1.59x | 1.78x |
| | MTBench | 23.136 | 21.463 | 1.66x | 1.79x |
| Llama2 13B/JF 160M | C4 | 20.6 | 19.007 | 1.87x | 2.03x |
| | OAsst | 23.711 | 22.378 | 1.60x | 1.70x |
| | WikiText-2 | 26.222 | 24.579 | 1.45x | 1.54x |
| | MTBench | 23.458 | 22.199 | 1.64x | 1.73x |

core principles of our split computing paradigm—offloading partial decoding workloads from server to edge—naturally extend to models of any size through the draft-verify speculative decoding scheme. Our upcoming research will quantify these benefits across the full spectrum of model scales.

## B.2 Performance with Lighter GPUs

We use the RTX 4090 as representative of consumer-grade GPUs, now widespread in both edge-cloud providers and user devices. However, to demonstrate broader generalizability, we conducted additional experiments with lighter consumer-grade GPUs. We measured SpecEdge performance using the RTX 3060 Ti and the RTX 2080 Ti. Table 4 shows the inter token latency and throughput with target/draft model pairs configured as Qwen3-14B/1.7B and Qwen3-14B/0.6B. Compared to the server-only approach, SpecEdge still attains meaningful throughput improvements even with less powerful GPUs.

Table 4: Performance comparison of lighter edge GPUs.

| Target/Draft Model | Edge GPU | Peak FP16 TFLOPS | Memory Bandwidth (GB/s) | Inter token Latency (ms) | Server Throughput (tok/s) |
| --- | --- | --- | --- | --- | --- |
| Qwen3-14B/1.7B | RTX 3060 Ti | 16.20 | 448.0 | 36.818 | 50.297 |
| | RTX 2080 Ti | 26.90 | 616.0 | 34.409 | 54.657 |
| | Server-only (A100 40GB) | 312 | 1555 | 32.451 | 30.816 |
| Qwen3-14B/0.6B | RTX 3060 Ti | 16.20 | 448.0 | 36.818 | 56.135 |
| | RTX 2080 Ti | 26.90 | 616.0 | 34.409 | 54.657 |
| | Server-only (A100 40GB) | 312 | 1555 | 32.326 | 30.935 |

## B.3 Performance with Non-Tree-based Speculative Decoding Method

To demonstrate the versatility of our approach beyond tree-structured methods, we implemented the speculative decoding technique from [Leviathan et al., 2023], which uses a linear candidate sequence rather than exploring multiple branching paths. This implementation allows us to evaluate whether SpecEdge's core innovation—disaggregating drafting and verification between edge and server—generalizes effectively across different speculative decoding paradigms.

Tables 5 and 6 present the results of this evaluation, comparing SpecEdge against the conventional server-only deployment. The throughput measurements in Table 5 demonstrate SpecEdge's efficiency

Table 5: Server throughput and cost analysis on SpecEdge with non-tree based speculative decoding method.

| Target/Draft Model | Server Throughput (tokens/s) | | Cost Efficiency (1k tokens/$) | |
|---|---|---|---|---|
| | Server-only | **SpecEdge** | Server-only | **SpecEdge** |
| Vicuna 33B/SL 1.3B | 24.527 | 39.370 (1.61x) | 17.484 | 24.649 (1.41x) |
| Llama2 13B/TL 1.1B | 33.044 | 64.851 (1.96x) | 29.372 | 49.150 (1.67x) |
| Llama2 13B/JF 160M | 32.755 | 49.639 (1.52x) | 29.116 | 37.621 (1.29x) |

Table 6: Inter token latency (ITL) comparison on SpecEdge with non-tree based speculative decoding method.

| Target/Draft Model | ITL (ms) | | Speedup | |
|---|---|---|---|---|
| | Server-only | **SpecEdge** | Server-only | **SpecEdge** |
| Vicuna 33B/SL 1.3B | 40.771 | 38.067 | 1.61x | 1.72x |
| Llama2 13B/TL 1.1B | 30.263 | 26.085 | 1.40x | 1.62x |
| Llama2 13B/JF 160M | 30.530 | 30.818 | 1.38x | 1.37x |

advantages, while Table 6 quantifies the relative speedup over server-only autoregressive decoding. These results confirm that our disaggregated architecture delivers consistent improvements regardless of the underlying speculative decoding strategy, reinforcing the broad applicability of our approach.

## B.4 Performance with Batch Drafting Method

While our main deployment architecture assumes that each concurrent request utilizes a dedicated edge GPU, we investigate an alternative architecture where a single edge GPU generates draft tokens for multiple concurrent requests. This alternative method offers trade-off for scenarios where operators want to utilize fewer consumer-grade edge GPUs. We evaluate this configuration by batching multiple requests on the RTX 4090 drafter with batch sizes from 2 to 4, measuring both inter token latency and cost efficiency across different target/draft model pairs.

The results show a trade-off between cost efficiency and latency. As shown in Table 7, cost efficiency improves with the alternative method, ranging from 4.4% to 29.5% better across different configurations due to better edge GPU utilization. However, this comes at the expense of increased inter token latency (5.9% to 19.0% slower), primarily caused by contention from batching multiple requests and longer draft-to-verify cycles. This alternative deployment method could be preferable in cost-sensitive scenarios where higher latency is acceptable.

Table 7: Performance comparison of alternative deployment method.

| Target/Draft Model | Batch Size | Inter Token Latency (ms) | Cost Efficiency (1k toks/$) |
|---|---|---|---|
| | 2 | 31.777 (6.8% slower) | 67.299 (14.7% better) |
| Qwen3-14B/1.7B | 3 | 36.617 (13.0% slower) | 71.369 (29.5% better) |
| | 4 | 40.099 (16.7% slower) | 85.662 (24.5% better) |
| | 2 | 30.494 (8.5% slower) | 67.680 (5.8% better) |
| Qwen3-14B/0.6B | 3 | 34.028 (15.2% slower) | 80.252 (12.0% better) |
| | 4 | 39.035 (19.0% slower) | 82.376 (20.0% better) |
| | 2 | 43.539 (5.9% slower) | 51.245 (4.4% better) |
| Qwen3-32B/1.7B | 3 | 48.889 (3.9% slower) | 59.635 (21.3% better) |
| | 4 | 58.402 (9.5% slower) | 57.857 (25.1% better) |

### B.5 Performance under Reasoning Mode

Many modern LLMs support reasoning capabilities to enhance their inference performance. Reasoning tokens often exhibit highly repetitive patterns, which can impact the acceptance rate in speculative decoding. To investigate the inference performance with reasoning capabilities, we measured the accepted tokens, server throughput, and cost efficiency with and without reasoning mode enabled.

Table 8 presents the accepted tokens, server throughput, and cost efficiency for each target-draft model pair. When generating reasoning tokens, we observe consistent improvements across all three metrics: accepted tokens, server throughput, and cost efficiency. These results indicate that SpecEdge can inherently benefit from improved speculative decoding performance coming from redundancy in reasoning processes.

Table 8: Performance comparison of SpecEdge with and without reasoning mode.

|  | Accepted Tokens | | Server Throughput (tok/s) | | Cost Efficiency (1k toks/$) | |
| --- | --- | --- | --- | --- | --- | --- |
| Target/Draft Model | Non-Reasoning | Reasoning | Non-Reasoning | Reasoning | Non-Reasoning | Reasoning |
| Qwen3-14B/1.7B | $3.80 \pm 1.54$ | $4.17 \pm 1.39$ | 64.343 | 72.249 | 48.883 | 54.898 |
| Qwen3-14B/0.6B | $3.68 \pm 1.50$ | $4.10 \pm 1.36$ | 70.703 | 81.890 | 53.693 | 62.157 |
| Qwen3-32B/1.7B | $4.09 \pm 1.95$ | $4.62 \pm 1.90$ | 24.880 | 27.617 | 39.430 | 43.373 |

## C  Cost Efficiency with Various GPU Providers and Cloud Service Providers

We anchored our cost estimates to widely available public pricing: A100 GPUs from Google Cloud Platform and RTX 4090 GPUs from Vast.ai [Vas, 2025]. To validate the robustness of our cost-efficiency claims, we conducted additional experiments across multiple cloud environments, including various GPU providers (Vast.ai, Runpod [run, 2025], and TensorDock [Ten, 2025]) and major cloud service providers (Google Cloud Platform [GCP, 2025], Amazon Web Services [AWS, 2025], and Microsoft Azure [Azu, 2025]). Table 9 and 10 show the GPU pricing from different providers. Table 11 demonstrates that SpecEdge maintains cost efficiency improvements consistently across all tested configurations.

Table 9: Edge GPU Pricing.

| GPU | GPU Provider | Cost ($/hr) |
| --- | --- | --- |
| RTX 4090 | RunPod | 0.69 |
| | Vast.ai | 0.35 |
| | TensorDock | 0.359 |
| RTX Pro 6000 | Vast.ai | 1.08 |
| | TensorDock | 1.15 |

Table 10: Server GPU Pricing.

| GPU | Cloud Service Provider | Cost ($/hr) |
| --- | --- | --- |
| A100 40GB | Google Cloud Platform | 4.05 |
| | Amazon Web Services | 4.10 |
| A100 80GB | Google Cloud Platform | 5.05 |
| | Amazon Web Services | 5.12 |
| | Microsoft Azure | 3.673 |

## D  Case Study: Proactive Edge Drafting in Action

This section illustrates SpecEdge's proactive drafting mechanism through an illustrative example with a sample query. We demonstrate how the system handles the complete drafting lifecycle: initial

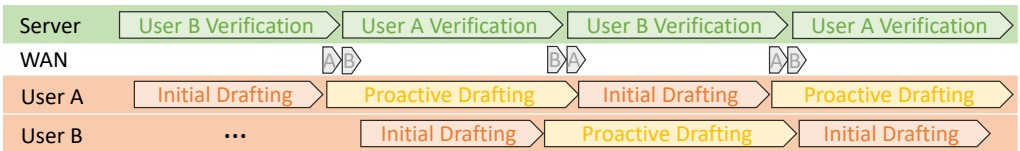

Figure 13: Timeline view showing parallel operations at edge and server, with proactive drafting occurring during server verification.

Table 11: Cost efficiency comparison of GPU Providers and Cloud Service Providers.

| Target/Draft Model | Server/Edge GPU | GPU Provider | Cost Efficiency (1k toks/$) | | |
| --- | --- | --- | --- | --- | --- |
| | | | Google Cloud Platform | Amazon Web Service | Microsoft Azure |
| Qwen3-14B/1.7B | A100 40GB / RTX 4090 | Vast.ai | 51.018 (1.87×) | 50.485 (1.87×) | - |
| | | RunPod | 44.721 (1.64×) | 44.311 (1.65×) | - |
| | | TensorDock | 50.829 (1.87×) | 50.30 (1.87×) | - |
| | | Baseline (Server-only) | 27.222 | 26.890 | - |
| Qwen3-14B/1.7B | A100 40GB / RTX Pro 6000 | Vast.ai | 47.68 (1.75×) | 47.30 (1.76×) | - |
| | | TensorDock | 46.64 (1.71×) | 46.27 (1.72×) | - |
| | | Baseline (Server-only) | 27.222 | 26.890 | - |
| Qwen3-14B/0.6B | A100 40GB / RTX 4090 | Vast.ai | 52.938 (1.86×) | 52.386 (1.88×) | - |
| | | RunPod | 46.382 (1.63×) | 45.957 (1.65×) | - |
| | | TensorDock | 52.741 (1.86×) | 52.192 (1.88×) | - |
| | | Baseline (Server-only) | 28.415 | 27.802 | - |
| Qwen3-32B/1.7B | A100 80GB / RTX 4090 | Vast.ai | 31.619 (1.82×) | 31.332 (1.83×) | 41.757 (1.75×) |
| | | RunPod | 28.453 (1.64×) | 28.144 (1.65×) | 36.280 (1.52×) |
| | | TensorDock | 31.619 (1.82×) | 31.239 (1.83×) | 41.591 (1.75×) |
| | | Baseline (Server-only) | 17.330 | 17.090 | 23.822 |

draft generation, proactive expansion, server verification, and subsequent drafting. Figure 13 shows the timeline view of each operation.

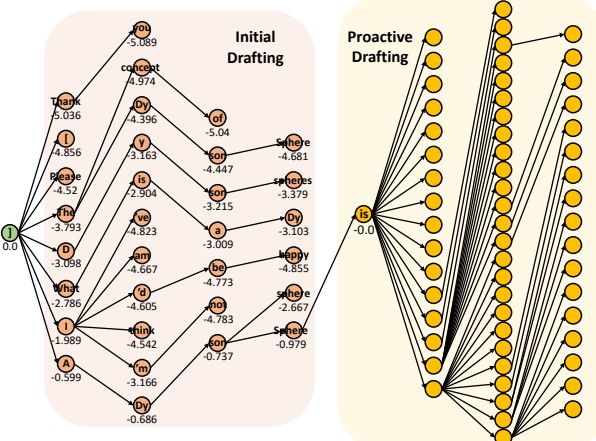

Figure 14: Example of proactive draft tree expansion after initial drafting. After creating the initial draft tree (left), the edge device identifies the most probable leaf token and proactively generates additional tokens (right) while awaiting server verification.

**Experimental Setup.** For this demonstration, we use a sample from the OAsst dataset with the query "What is a Dyson Sphere?". We employ Llama-3.2-3B [Grattafiori et al., 2024] as the target model and Llama-3.2-1B as the draft model.

**Initial Draft and Proactive Expansion.** Figure 14 illustrates the transition from initial drafting to proactive expansion. Once the edge device constructs the initial draft tree, it submits this tree to the server for verification. Simultaneously—rather than remaining idle—the edge identifies the expansion head with the highest cumulative log probability and begins generating additional tokens proactively.

**Server Verification and Alignment.** When the verification results arrive from the server, the edge device compares its locally expanded tree with these results. Figure 15 demonstrates a case of complete draft alignment, where the server's verified tokens ("A Dyson Sphere is") match both the initial draft tree path and the selected expansion head.

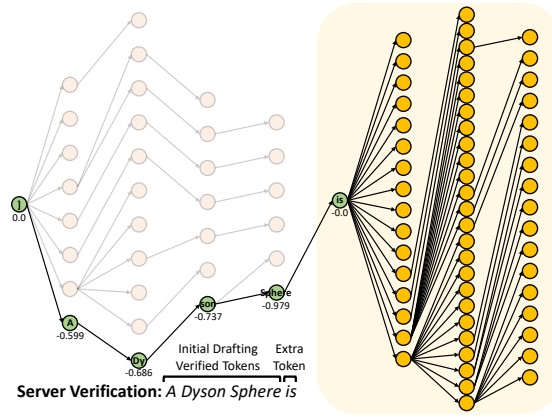 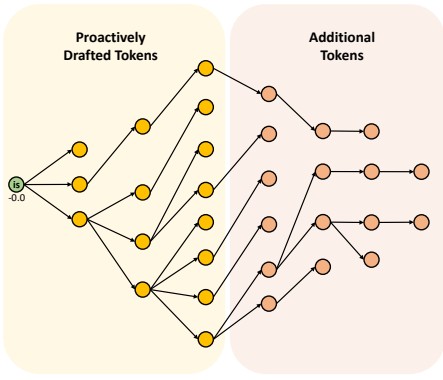

Figure 15: Server verification and complete draft align-ment. The server verification results (showing "A Dyson Sphere is") perfectly align with a path in the draft tree, validating both the initial draft and the chosen expan-sion head.

Figure 16: Deeper token drafting with proac-tively drafted tokens. The previously gener-ated proactive tokens tree following the extra token ("is") can be immediately used in the next drafting cycle without additional com-putation.

**Accelerated Token Drafting.** Following successful verification, SpecEdge leverages the proactively drafted tokens to benefit the next drafting cycle. Figure 16 shows how these pre-generated tokens contribute to the next draft submission, eliminating the need to regenerate these tokens and thus building deeper draft candidates.

**Performance Implications.** This example demonstrates how SpecEdge's proactive drafting strategy provides tangible performance benefits:

- **Reduced idle time:** The edge GPU remains productive during server verification periods.
- **Deeper subsequent drafting:** Pre-generated tokens allow additional draft forward passes for the next drafting cycle.
- **Effective resource utilization:** Computational resources on both edge and server are maximized.

The complete alignment case shown here represents the optimal scenario, though SpecEdge handles partial or misaligned drafts as well (Section 4.2).

