# OpenReview forum: "SpecEdge: Scalable Edge-Assisted Serving Framework for Interactive LLMs"
_NeurIPS.cc/2025/Conference — NeurIPS 2025 spotlight_

### Official Review · Reviewer_F335 · 2025-07-02

**Clarity:** 3
**Significance:** 2
**Originality:** 3
**Rating:** 4
**Confidence:** 4

**Summary:**

This paper introduces SpecEdge, a collaborative framework for serving large transformer models efficiently. The key insight is that by using consumer GPUs in edge devices to draft tokens during speculative decoding while using server hardware exclusively for verification, it is possible to reduce the Time-Between-Tokens while also being more cost-effective. This disaggregated serving-setup is evaluated on a range of model sizes and tasks using a combination of datacenter-grade GPUs (A100s) and a range of consumer-grade GPUs. Combining disaggregated deployment with pipeline-aware scheduling leads to considerable efficiency gains in both performance and cost as demonstrated by the evaluation. These improvements are despite the latency penalties incurred from communicating over a wide area network.

**Questions:**

1. Is it possible to provide some rough estimates on the minimum hardware required to still retain a competitive advantage over running server-only? All the hardware choices used in the evaluation have considerable VRAM/device memory and compute capability. A more compelling evaluation will include the scenario where the edge-devices are far lighter such as the integrated-GPUs on desktop chips or even mobile devices.


2. How will the cost analysis change if we are to run the server-only configuration at the maximum possible batch size? Since there is such an emphasis on cost efficiency, one can argue that we will almost always pick the most cost-effective option regardless of latency.


3. Related to the previous question, the evaluation assumes that every user has a consumer-grade GPU. Would it be even more cost-effective to run the system with a single RTX-4090 running drafts on all requests while the A100 executes the primary model? This does go against some of the key arguments the paper makes about the prevalence of capable edge-devices but it will be interesting to see if this option works out better.


4. The paper states that the Prefill stage takes an identical amount of time on both the server and the edge. Wouldn’t the more accurate statement be: prefill_time = max(server_time, edge_time)?

**Ethical Concerns:**

["NO or VERY MINOR ethics concerns only"]

**Final Justification:**

The primary reason for the current rating is that the technique needs hardware that consumes a lot of power to be effective. This is despite purportedly being an edge device technique. As such, while sound, it's not necessarily very exciting.

**Limitations:**

No, the paper can be a little more forthcoming about the precise compute capability of the edge device necessary for SpecEdge to provide an advantage.

**Quality:**

3

**Strengths And Weaknesses:**

**Strengths:**

1. The system has many well-designed components including pipeline-aware scheduling. This scheduling is necessary for SpecEdge to make a difference in practical settings by avoiding idle time.

2. The reduction in Time-Between-Tokens is compelling under the given experimental setting which includes a range of model sizes and tasks.

3. Interesting insight about doubling-down on the most likely path while picking which node to continue on while proactively drafting.

**Weaknesses:**

1. The evaluation stretches the definition of what can be deemed an edge device. Gains will diminish with less-performant hardware that is much more prevalent compared to the expensive consumer-grade GPUs used in the evaluation.

2. While a cost analysis takes center stage in the evaluation, it is unclear if the cost numbers used  are sound. In particular, using the cost of renting an edge GPU isn’t equivalent to owning one (which might actually be cheaper when amortized over duration of ownership).

---

> ### Author Rebuttal · Authors · 2025-07-31
>
> We thank the reviewer for recognizing SpecEdge’s design and efficiency gains.
>
> **On "edge" hardware definition and lighter devices:**
>
> We use RTX 4090s as representative of high-end consumer GPUs, now widespread in both edge-cloud providers and user devices [Valve HW survey, 2024]. SpecEdge remains effective on older or lighter GPUs: Fig. 12 shows speedup gains even with RTX 2080 Ti compared to autoregressive.
> To address the minimum hardware requirements, we provide additional analysis on mid-range and lower-end devices. Our current implementation requires 6.5GB VRAM for Qwen3-1.7B FP16 draft model and 4GB for Qwen3-0.6B FP16 draft model (sequence length 2048) due to CUDA Graph usage for reducing kernel launch overhead and lack of optimizations like PagedAttention for KV cache management.
> In latency, based on our measurements, SpecEdge maintains competitive advantage (latency on-par or better than server-only) with RTX 2080Ti (34.4ms vs 32.5ms server-only) and RTX 3060Ti (36.8ms vs 32.5ms server-only), representing a reasonable threshold for interactive applications. While it's challenging to establish a clear lower bound between FLOPS and memory bandwidth due to complex factors like tree reordering in draft expansion and tree attention forward passes, devices with specifications similar to RTX 2080 Ti can be considered the current lower threshold for achieving latency parity with server-only implementations in our current system.
>
> | Target/Draft   | Device                  | Peak FP16 TFLOPS | Memory Bandwidth (GB/s) | Inter-token latency (ms) | Throughput (tok/s) |
> | -------------- | ----------------------- | ---------------- | ----------------------- | ------------------------ | ------------------ |
> | Qwen3-14B/1.7B | RTX 3060 Ti             | 16.20            | 448.0                   | 36.818                   | 50.297             |
> |                | RTX 2080 Ti             | 26.90            | 616.0                   | 34.409                   | 54.657             |
> |                | Server Only (A100 40GB) | 312              | 1555                    | 32.451                   | 30.816             |
> | Qwen3-14B/0.6B | RTX 3060 Ti             | 16.20            | 448.0                   | 33.801                   | 56.135             |
> |                | RTX 2080 Ti             | 26.90            | 616.0                   | 32.562                   | 58.415             |
> |                | Server Only (A100 40GB) | 312              | 1555                    | 32.326                   | 30.935             |
>
> For our fine-grained analysis, we consider the Qwen3-14B/Qwen3-0.6B as a target/draft model pair. In our evaluation, this model pair achieves an inter token latency of 32.326 ms/token in the server-only scenario. To achieve comparable latency in SpecEdge using specific device, we must consider various tree depths to utilize GPU capabilities. For simplicity in our analysis, we assume that draft time is always less than or equal to verification time. Under this assumption, SpecEdge’s user perceived latency can be expressed as: (server time x 2) / (generated tokens per verification). Therefore, the lowest bound of mean draft step latency according to beam length would be below table. In this criterion, we can choose appropriate device roughly using mean draft step latency metrics.
>
> | Beam length | Gen. token per verify | Mean draft step latency |
> | ----------- | --------------------- | ----------------------- |
> | 3           | 3.15                  | 13.07                   |
> | 4           | 3.54                  | 8.726                   |
> | 5           | 3.96                  | 6.240                   |
> | 6           | 4.18                  | 4.926                   |
>
> For non-CUDA platforms like M4 and CPU-based solutions, our current implementation does not support them due to time limitations. However, current NPU in SoC can support viable speed for SpecEdge Draft process like Apple M-Chips (150 tokens/s on iPhone Pro 16, Qwen 0.5B INT4)[2], Snapdragon Chips (68.2 tokens/s on Snapdragon 8 Elite Mobile, PLaMo-1B w4a16) [3] . While we cannot implement optimization for various platforms, edge resources can be the plausible edge resources in cost-sensitive situations.
>
> **Server-only performance at maximum batch size:**
>
> Running server-only configurations at maximum batch size would indeed improve cost efficiency by better amortizing model parameter memory access across denser GPU computations. However, this approach fundamentally conflicts with the latency requirements of interactive LLM applications, where users expect responsive token generation rather than batch-optimized throughput. We briefly conducted an experiment with Qwen3-14B/Qwen3-0.6B where the server batch size was 6.
>
> |             | Accepted Tokens | Inter-token Latency | Server Throughput (token/s) | Cost Efficiency (1k toks/$) |
> | ----------- | --------------- | ------------------- | --------------------------- | --------------------------- |
> | SpecEdge    | 3.93±1.63       | 55.764              | 209.20                      | 91.768                      |
> | Server-only | 3.2±1.16        | 56.499              | 62.603                      | 87.002                      |
>
> While maximum batch size configurations optimize for cost efficiency, they sacrifice the responsiveness essential for interactive applications. SpecEdge addresses this fundamental tension by maintaining both cost efficiency and low latency, making it suitable for interactive scenarios where pure batch processing approaches would be inadequate. For applications where latency is less critical, SpecEdge's cost advantages would be even more pronounced in large batch and long-context scenarios.
>
> **On alternative deployment models:**
>
> We include experiments for inter-token latency and cost-efficiency where a single RTX 4090 drafts for multiple requests.
>
> | Model Pair         | Batch Size | Inter Token Latency (ms) | Cost Efficiency<br>(1k toks/$) |
> | ------------------ | ---------- | ------------------------ | ------------------------------ |
> | **Qwen3 14B/1.7B** | 2          | 31.777 (6.8% slower)     | 67.299 (14.7% better)          |
> |                    | 3          | 36.617 (13.0% slower)    | 71.369 (29.5% better)          |
> |                    | 4          | 40.099 (16.7% slower)    | 85.662 (24.5% better)          |
> | **Qwen3 14B/0.6B** | 2          | 30.494 (8.5% slower)     | 67.680 (5.8% better)           |
> |                    | 3          | 34.028 (15.2% slower)    | 80.252 (12.0% better)          |
> |                    | 4          | 39.035 (19.0% slower)    | 82.376 (20.0% better)          |
> | **Qwen3 32B/1.7B** | 2          | 43.539 (5.9% slower)     | 51.245 (4.4% better)           |
> |                    | 3          | 48.889 (3.9% slower)     | 59.635 (21.3% better)          |
> |                    | 4          | 58.402 (9.5% slower)     | 57.857 (25.1% better)          |
>
> This approach further reduces the per-token cost, but it comes at the expense of higher user-perceived latency due to contention and longer draft-to-verify cycles.
>
> **On prefill-time description:**
>
> We agree and will correct the statement to “prefill_time = max(server_time, edge_time).” We will incorporate these clarifications in the revision. Note that, prefill time of server time was always bigger than prefill time of edge throughout our experiment settings.
>
> [1] Sadhukhan, Ranajoy, Jian Chen, Zhuoming Chen, Vashisth Tiwari, Ruihang Lai, Jinyuan Shi, Ian En-Hsu Yen, Avner May, Tianqi Chen, and Beidi Chen. "Magicdec: Breaking the latency-throughput tradeoff for long context generation with speculative decoding." arXiv preprint arXiv:2408.11049 (2024).
>
> [2] S. Willison. Run LLMs on macOS using llm-mlx and Apple's MLX framework. Simon Willison's Weblog, Feb. 2025. https://simonwillison.net/2025/Feb/15/llm-mlx/
>
> [3] Preferred Networks. PLaMo-1B. Qualcomm AI Hub, 2025. https://aihub.qualcomm.com/models/plamo_1b

---

> > ### Comment · Area_Chair_QXJt · 2025-08-05
> > **Dear reviewer, please read the rebuttal and participate in discussions with authors. Thank you**
> >
> > Dear reviewer, please read the rebuttal and participate in discussions with authors. Thank you

---

> > ### Comment · Reviewer_F335 · 2025-08-06
> >
> > Thank you for providing more results and estimating what the minimum hardware specs need to be to still see latency benefits. I completely understand why evaluating on non-CUDA platforms will be cumbersome. I will maintain the current rating.

---

### Official Review · Reviewer_dKE8 · 2025-07-02

**Clarity:** 3
**Significance:** 2
**Originality:** 3
**Rating:** 4
**Confidence:** 3

**Summary:**

This paper introduces SpecEdge, a novel edge-assisted inference framework designed to reduce the cost and improve the efficiency of serving Large Language Models (LLMs) by leveraging consumer-grade GPUs at the network edge. SpecEdge proposes a speculative decoding scheme that splits LLM workloads: edge GPUs handle token drafting, while server GPUs perform final verification. This approach minimizes network communication by exchanging only token outputs, rather than full model states needed in TP or PP. SpecEdge employs two key techniques: "Proactive Edge Drafting," which allows edge GPUs to continue generating tokens while awaiting server verification, effectively masking network latency, and "Server-side Pipeline-aware Scheduling," which interleaves multiple user requests to maintain high server GPU utilization. Experiments demonstrate that SpecEdge significantly enhances overall cost efficiency (1.91x), increases server-side throughput (2.22x), and reduces inter-token latency (11.24%) compared to server-only baselines, even under realistic wide-area network conditions.

**Questions:**

1. In a distributed system spanning edge and cloud, what mechanisms does SpecEdge employ or envision for handling edge device failures, network interruptions to ensure continuous and reliable LLM serving?
2. While SpecBench covers six tasks, the evaluation primarily uses a batch size of 1, appropriate for interactive LLM serving. A more extensive analysis of how SpecEdge performs under varying and dynamic workloads (e.g., larger batch sizes, different request arrival patterns, mixed interactive/batch workloads) would strengthen its generalizability.

**Ethical Concerns:**

["NO or VERY MINOR ethics concerns only"]

**Final Justification:**

The response addresses part of my concern. I will keep my scores.

**Limitations:**

yes

**Quality:**

2

**Strengths And Weaknesses:**

Strengths
1. Timely and important problem.
2. Novel way of splitting LLM inference workload -- Disaggregate Token Drafting (Edge) and Verification (Server) to minimize communication cost WANs.
3. Good performance -- SpecEdge can enhance overall cost efficiency (1.91x), increase server-side throughput (2.22x), and reduce inter-token latency (11.24%) compared to server-only baselines.
4. Comprehensive Evaluation and Analysis: The evaluation covers various LLM models, draft models, and datasets.

Weaknesses:
1. The framework assumes widespread availability and reliable operation of consumer-grade GPUs at the edge. While this is increasingly true, the paper doesn't delve into the complexities of managing a large, potentially heterogeneous, and less controlled fleet of edge devices (e.g., fault tolerance, security, user-provided hardware), which makes the practical use of the proposed method a big concern.

---

> ### Author Rebuttal · Authors · 2025-07-31
>
> We thank the reviewer for highlighting the importance of our problem and the strengths of our evaluation.
>
> **On reliability under edge failures:**
>
> SpecEdge's speculative decoding architecture provides inherent resilience through its draft-and-verify paradigm. In case of edge device failures or network interruptions, the system can utilize other available edge cloud GPUs as fallback resources.
> When edge assistance becomes unavailable, the service providers could make SpecEdge gracefully falls back to server-only speculative decoding or standard autoregressive generation using the most recent verified tokens, ensuring uninterrupted service. Also, The service operator could implement dynamic draft depth adjustment mechanism to adapt real-time edge device performance degradation by reducing speculation depth or redistributing load among healthy edge nodes. We will discuss these fault-tolerance mechanisms and failure handling strategies in detail in our revision.
>
> **On heterogeneous and less-controlled edge devices:**
>
> We appreciate the reviewer's important observation regarding operational complexities of managing heterogeneous edge fleets. SpecEdge addresses some heterogeneity through dynamic draft depth adjustment (§4.3), which adapts to varying edge device capabilities in real-time and supports mixed consumer-grade GPU configurations.
> We acknowledge that comprehensive edge fleet management entails additional challenges including fault tolerance, security, dynamic resource scaling, and load balancing. We believe many of these operational aspects can be addressed through existing distributed serving systems. For example, Kubernetes-based LLM serving platforms like llm-d and AIBrix already provide robust scheduling, load balancing, and fault tolerance mechanisms for distributed inference workloads. SpecEdge's architecture is similar to microservice architecture, with our implementation using gRPC communication between the drafter and verifier components, making it naturally compatible with existing distributed system management frameworks.
> Our work addresses the key algorithmic challenge of enabling collaborative LLM serving through speculative execution on resource-constrained edge devices. While optimization and deployment at scale present interesting research challenges that we are open to exploring in future work, we believe these can be effectively tackled through integration with mature orchestration platforms.
>
> **On evaluation under varying and dynamic workloads:**
>
> We appreciate the reviewer's observation regarding our evaluation scope. While our primary results focus on batch size 1 for latency-sensitive interactive serving, our experiments do include evaluation across different batch sizes. Figure 6 demonstrates SpecEdge's throughput and latency advantages scaling across batch sizes from 1 to 4, showing that our approach maintains benefits even beyond the interactive setting where batch size 1 is most critical.
>
> We acknowledge that exploring dynamic request arrival patterns and mixed interactive/batch workloads would strengthen the generalizability analysis. Handling variable arrival patterns primarily involves load balancing and resource scaling mechanisms that can be implemented using existing distributed serving infrastructure (as discussed in our edge fleet management response). We plan to work on Kubernetes-based orchestration to demonstrate SpecEdge's benefits under more dynamic workloads and include it in the near future.

---

> > ### Comment · Reviewer_dKE8 · 2025-08-06
> >
> > Thanks for your responses. I will keep my scores.

---

> ### Comment · Area_Chair_QXJt · 2025-08-05
> **Please participate in discussions with authors**
>
> Dear reviewer, please read the rebuttal and participate in discussions with authors. Thank you

---

### Official Review · Reviewer_wvG4 · 2025-07-03

**Clarity:** 2
**Significance:** 3
**Originality:** 3
**Rating:** 5
**Confidence:** 4

**Summary:**

The authors proposed a cost-efficient approach for LLM inference by implementing an Edge-Cloud coordinated speculative sampling. Via doing draft proposing on the edge devices, the computational load on the cloud side can be largely alleviated. Building upon this novel workflow, this paper also refined the drafting strategy by **proactive drafting** to gain advantage before the verification can be done.
Kill a granny

**Questions:**

- Is there any comparison based on SOTA LLM serving frameworks like SGLang or vLLM?
- In edge settings of Table 1, the actual computation resource is more abundant compare to server-only setting. Like you've mentioned earlier, A100 is nealy the same FLOPs tensor performance as 4090. I consider this not an apple-to-apple comparison. Do you have any idea to recitify this concern?

**Ethical Concerns:**

["NO or VERY MINOR ethics concerns only"]

**Final Justification:**

The rebuttal addressed my concerns.

**Limitations:**

Yes

**Quality:**

3

**Strengths And Weaknesses:**

## Strengths
- The motivation of offload computational pressure to edge devices is very solid, and speculative sampling is indeed a good solution without heavy communication overhead.
- The proposed proactive sampling strategy well complements the round trip latency introduce in edge-cloud scene.
- Evaluation of SpecEdge is very thorough and comprehensive.

## Weakness
- The cost estimation is concerning throughout the paper. Information is missing explaining how did the authors come to the comparison. Generation cost can largely vary from the cloud providers, infrastructures, server loads, and implementations. For instance, MaaS providers has orders lower pricing for mentioned models. The mentioned cost estimation merely base on one quote from a single provider for time-sliced GPU usage is not pragmatically informative enough, at least ignores depreciation, power and DevOps overheads; comparative TCO is therefore uncertain.
- From my understanding, SpecEdge inherently introduces tradeoff between two key SLOs: time-to-first-token (TTFT) and time-per-output-token (TPOT). This paper focused on TPOT, namely "inter-token latency" in the paper, but not TTFT.
- Some minor grammar issue like l98, page 3 "is split"

---

> ### Author Rebuttal · Authors · 2025-07-31
>
> We thank the reviewer for recognizing the motivation, and thoroughness of our work.
>
> **On cost estimation and analysis:**
>
> We acknowledge the reviewer's concerns. Our cost-efficiency metric reports tokens generated per dollar based on expected cloud bills rather than an absolute total cost of ownership (TCO). A full TCO analysis—including depreciation, power, and DevOps overhead—is indeed desirable but infeasible due to the lack of accessible, standardized data across providers. Instead, to keep the comparison reproducible, we anchored our estimates to the widely available pricing: A100 server GPUs from Google Cloud Platform and Vast.ai RTX 4090 consumer GPUs, two of the most widely used services in their respective market segments (hyperscale cloud and edge cloud). Although absolute numbers may vary across environments, our conclusions hold as long as the significant and well-established price gap between consumer-grade and server-grade GPUs persists, which is the case today.
>
> We included performance analysis in various cloud providers, including three major cloud providers and three edge GPU providers. As shown in the table below, our approach maintains substantial cost efficiency improvements (1.5-1.9x) across various configurations.
>
> - RTX 4090
>   - RunPod: 0.69 $/hr
>   - Vast.ai: 0.35 $/hr
>   - TensorDock: 0.359 $/hr
> - RTX Pro 6000
>   - Vast.ai: 1.08 $/hr
>   - TensorDock: 1.15 $/hr
> - A100 40GB
>   - Google Cloud Platform: 4.05 $/hr
>   - Amazon Web Service: 4.10 $/hr
>   - Microsoft Azure does not provide A100 40GB currently.
> - A100 80GB
>   - Google Cloud Platform: 5.05 $/hr
>   - Amazon Web Service: 5.12 $/hr
>   - Microsoft Azure: 3.673 $/hr
>
> Qwen3-14B / Qwen3-1.7B
> (1k toks/$)
>
> | GPU Provider \ CSP     | Google Cloud Platform | Amazon Web Service |
> | ---------------------- | --------------------- | ------------------ |
> | Vast.ai                | 51.018 (1.87x)        | 50.485 (1.87x)     |
> | RunPod                 | 44.721 (1.64x)        | 44.311 (1.65x)     |
> | TensorDock             | 50.829 (1.87x)        | 50.30 (1.87x)      |
> | Baseline (Server-only) | 27.222                | 26.890             |
>
> Qwen3-14B / Qwen3-0.6B
> (1k toks/$)
>
> | GPU Provider \ CSP     | Google Cloud Platform | Amazon Web Service |
> | ---------------------- | --------------------- | ------------------ |
> | Vast.ai                | 52.938 (1.86x)        | 52.386 (1.88x)     |
> | RunPod                 | 46.382 (1.63x)        | 45.957 (1.65x)     |
> | TensorDock             | 52.741 (1.86x)        | 52.192 (1.88x)     |
> | Baseline (Server-only) | 28.415                | 27.802             |
>
> Qwen3-32B / Qwen3-1.7B
> (1k toks/$)
>
> | GPU Provider \ CSP     | Google Cloud Platform | Amazon Web Service | Microsoft Azure |
> | ---------------------- | --------------------- | ------------------ | --------------- |
> | Vast.ai                | 31.619 (1.82x)        | 31.332 (1.83x)     | 41.757 (1.75x)  |
> | RunPod                 | 28.453 (1.64x)        | 28.144 (1.65x)     | 36.280 (1.52x)  |
> | TensorDock             | 31.619 (1.82x)        | 31.239 (1.83x)     | 41.591 (1.75x)  |
> | Baseline (Server-only) | 17.33                 | 17.09              | 23.822          |
>
> Qwen3-14B / Qwen3-1.7B using RTX Pro 6000 as edge
>
> | GPU Provider \ CSP     | Google Cloud Platform | Amazon Web Service |
> | ---------------------- | --------------------- | ------------------ |
> | Vast.ai                | 47.68 (1.75x)         | 47.3 (1.76x)       |
> | TensorDock             | 46.64 (1.71x)         | 46.27 (1.72x)      |
> | Baseline (Server-only) | 27.222                | 26.890             |
>
> **On TTFT vs. TPOT:**
>
> There is no trade-off between time-to-first-token (TTFT) and time-per-output-token (TPOT) in SpecEdge. TTFT is dominated by the target model’s prefill stage, which is identical for both server-only (Autoregressive and Speculative Decoding) and SpecEdge configurations, while SpecEdge improves TPOT by reducing inter-token latency through proactive drafting and pipeline-aware scheduling. Following Reviewer F335’s feedback, we will clarify that the effective prefill latency is max(server_time, edge_time). During our evaluation, the prefill of the target model was always bigger than the draft model, so TTFT remained unchanged. but we will clarify the description and update in the revision.
>
> **On comparisons with SOTA serving systems:**
>
>  SpecEdge is orthogonal toward frameworks such as vLLM and SGLang; they focus on memory optimization and scheduling within the server, whereas SpecEdge targets cross-device disaggregation. These frameworks already support existing speculative decoding methods like Eagle and Medusa, demonstrating their compatibility with speculative approaches. We believe SpecEdge can similarly be integrated with these systems to inherit their server-side optimizations during the verification phase while adding the novel capability of edge-assisted drafting.
>
> **On Table 1 comparisons:**
>
> We appreciate the reviewer's observation regarding computational resources in our edge versus server-only settings. While we acknowledge that aggregate tensor FLOPS may differ between configurations, we respectfully argue that a direct FLOPS-to-FLOPS comparison does not reflect the practical realities of LLM serving.
>
> While tensor FLOPS is important for LLM serving, consumer-grade GPUs face fundamental constraints that prevent direct large model deployment. Although consumer-grade GPUs are approximately 10x less expensive and effective for serving smaller language models, the RTX 4090 has critical limitations: 24GB VRAM versus A100's 80GB, and approximately 1TB/s memory bandwidth versus A100's 2TB/s. These constraints make it impossible to serve large models like our 32B LLM on individual edge devices, regardless of theoretical compute capacity. Furthermore, traditional distributed serving approaches require high-bandwidth interconnects (NVLink, InfiniBand) that are absent in consumer environments, making conventional tensor/pipeline parallelism impractical for edge deployment.
>
> This is precisely why edge GPUs remain more cost-effective per FLOP yet largely unusable for large LLM serving—a paradox our work directly addresses. SpecEdge's core contribution is enabling collaborative serving of large LLMs using these memory-constrained, bandwidth-limited edge resources that were previously unsuitable for such workloads. Our approach represents a paradigm shift from server-centric to edge-assisted serving, unlocking vast amounts of distributed edge capacity rather than directly replacing server hardware.
>
> Given our design philosophy of leveraging the cost advantages of edge resources in practical deployments, cost-normalized evaluation (tokens/dollar) more accurately represents real-world utility. When we report server throughput (tokens/s), we aim to expose underlying capacity gains, not to imply a FLOPS-equivalent comparison.
> We will clarify this and evaluation rationale in our revision to ensure readers understand that we are demonstrating practical utilization of edge resources for tasks previously restricted to high-end server hardware, rather than claiming FLOPS-equivalent performance.

---

> > ### Comment · Reviewer_wvG4 · 2025-08-05
> > **On SOTA comparison**
> >
> > Thank you for your response. I understand SpecEdge is substantially distinct from those LLM serving frameworks. Apologies for not being clear. I meant that could some results be different if both client or server side use those frameworks? Optimizations in those backend can possibly hinder some overhead and results under them are more pragmatic.

---

> ### Author Response · Authors · 2025-08-05
> **Expected performance with SGLang integration**
>
> (Edited for additional details)
>
> We appreciate your engagement and thank you for clarifying things.
>
> We expect SGLang integration to reduce verification latency by 24.77% for the 32B model, 21.10% for the 14B model, and draft latency by 19.69% for the 1.7B model for the following reasons.
>
> The speculative decoding pipeline consists of two main phases. Draft-phase latency primarily includes (draft model forward + draft tree expansion) × tree depth, while verification-phase latency includes verify model forward, tree reordering, and KV-cache reordering (target and draft). We have already minimized tree expansion and reordering overhead and applied optimizations such as CUDA graph to reduce kernel launch overheads, making our current implementation performance comparable to state-of-the-art frameworks. Therefore, the expected gain from integrating SGLang would come mainly from further accelerating model forward latency compared to the default HuggingFace implementation.
>
> To estimate this impact, we benchmarked SGLang’s acceleration on model forward passes, observing 27.57% reduction for the 32B model, 29.36% for the 14B model, and 24.81% for the 1.7B model. These improvements were applied to the draft and verification phases to project the latency for both SpecEdge and the server-only speculative decoding baseline.
>
> Based on these latency estimates, we simulated throughput and cost efficiency:
>
> Table. Expected throughput and cost efficiency with SGLang acceleration.
>
> **Qwen3-14B/Qwen3-1.7B**
>
> |                 | Accepted Tokens | Server Throughput (token/s) | Cost Efficiency (1k toks/$) |
> | --------------- | --------------- | --------------------------- | --------------------------- |
> | SpecEdge        | 3.99            | 90.434 (2.463x)             | 61.306 (1.878x)             |
> | Server-only | 3.24            | 36.718                      | 32.638                      |
>
> **Qwen3-32B/Qwen3-1.7B**
>
> |             | Accepted Tokens | Server Throughput (token/s) | Cost Efficiency (1k toks/$) |
> | ----------- | --------------- | --------------------------- | --------------------------- |
> | SpecEdge    | 4.32            | 71.039 (2.490x)              | 42.420 (2.086x)             |
> | Server-only | 3.20             | 28.520                      | 20.331                      |
>
> Note that SGLang accelerates the verification phase more than the draft phase. With 4.3 Design component, SpecEdge interleaves verification requests at the server without idle time, resulting in (# Accepted Tokens) / (Verification Latency) as throughput. Meanwhile, server-only speculative decoding method results in (# Accepted Tokens) / (Draft Latency + Verification Latency) as throughput. Due to such dynamics, SpecEdge is expected to show greater gain when verification latency accelerates more than draft, and less otherwise.
>
> In conclusion, integrating SGLang improves absolute throughput and cost efficiency for both SpecEdge and server-only speculative decoding method while slightly increasing SpecEdge’s relative advantage.

---

> > ### Author Response · Authors · 2025-08-07
> >
> > We hope that the additional explanations and results we have provided have sufficiently addressed your original concerns and questions. If our clarifications have resolved the points you raised, we would be grateful if you would kindly consider reflecting this in your overall evaluation of the paper. We appreciate your review and engagement, which have helped us strengthen the analysis of our work.
> >
> > Thank you again for your time and feedback.

---

> > > ### Comment · Reviewer_wvG4 · 2025-08-08
> > > **Thanks**
> > >
> > > Dear Authors,
> > >
> > > Thank you for your response. I will adjust my rating accordingly.

---

### Official Review · Reviewer_ULXi · 2025-07-05

**Clarity:** 3
**Significance:** 4
**Originality:** 3
**Rating:** 5
**Confidence:** 4

**Summary:**

SpecEdge is a speculative decoding technique designed for efficient deployment across edge GPUs (like GTX 4090) and server GPUs (like A100). The core idea is to have edge GPUs generate speculative tokens, while server GPUs verify them. It introduces two key optimizations: Proactive Drafting and Dynamic Verification Scheduling. These strategies aim to reduce latency and enhance the efficiency of distributed speculative decoding.

**Questions:**

See above.

**Ethical Concerns:**

["NO or VERY MINOR ethics concerns only"]

**Final Justification:**

I think the authors address most of my concerns. I will keep my score.

**Quality:**

3

**Strengths And Weaknesses:**

Strengths:
- The idea of a heterogeneous edge-server serving is interesting and could be valuable for practical deployments. SpecEdge points in and demonstrates a promising direction. The new cost efficiency metric on the top of standard throughput or latency is very convincing.
- The evaluation is pretty thorough and up-to-date (impressive to even include Qwen3 which just came out). It includes a lot of standard opensource models and benchmarks.

Weaknesses and Questions:
- Ablations: SpecEdge uses specexec beam search tree construction and it's great to see the ablation of using tree or chain in appendix E (which I think could put in the main paper). Usually large tree constructions like specexec or sequoia would have a trade-off considering the hardware overhead for the tree shape (prefer wider than deeper since there's drafting cost). I'm curious to understand how specedge affect this? The distributed setting similar to offloading is perfect for allowing large tree (hide the overhead), does it allow deeper tree shape? I think it would be great to include a discussion tree construction in specedge setting although tree construction is based on prior work.
- More general settings: beside offloading and distributed setting, longcontext as analyzed in [1] is also a great setting for spec dec to work for large batches. Can you discuss how Specedge performs in such setting?
- I noticed it uses Qwen3 and Math pair for evaluation. Can you elaborate the acceptance rate for reasoning models for long generation tasks? I thought it would be higher since there're a lot of redundant generation sentences etc.

[1] Ranajoy Sadhukhan, Jian Chen, Zhuoming Chen, Vashisth Tiwari, Ruihang Lai, Jinyuan Shi, Ian En-Hsu Yen, Avner May, Tianqi Chen, Beidi Chen. MagicDec: Breaking the Latency-Throughput Tradeoff for Long Context Generation with Speculative Decoding.

---

> ### Author Rebuttal · Authors · 2025-07-31
>
> We thank the reviewer for the insightful comments.
>
> **On tree depth adjustment in SpecEdge:**
>
> SpecEdge leverages tree-based speculative decoding (from SpecExec) and dynamically adapts the draft tree depth to match the verification cycle, which includes both server-side verification and network RTT (4.3, Fig. 10). When the verification cycle is longer, SpecEdge can build deeper proactive draft trees by performing additional draft forward passes during the verification time, which increases the likelihood of more tokens being accepted per draft-verify cycle, while effectively hiding the additional drafting cost behind verification and communication. We will add a discussion about such a tree construction strategy in the revision.
>
> **On long-context and large-batch settings:**
>
> Like other speculative decoding approaches, SpecEdge achieves greater benefits in memory-bound scenarios. In short-context cases, widening or deepening the draft tree increases computational load and must be balanced against verification cost. In long-context scenarios, however, KV cache memory access dominates, making both deeper and wider trees feasible. This would allow SpecEdge to achieve higher acceptance rates, akin to the gains reported by MagicDec [1]. We would also add such expected behaviors in the revision.
>
> **On acceptance rates for reasoning models:**
>
> We thank the reviewer for the suggestion. In our main submission, we evaluated Qwen3 with reasoning capability turned off. We have since rerun experiments with reasoning mode enabled and observed that the average number of accepted tokens per verification cycle indeed increases, due to the redundancy in reasoning model outputs. We will include these quantitative results in the revision.
>
> | Target/Draft   | Reasoning     | Accept Tokens | Server Throughput (tok/s) | Cost Efficiency (1k toks/$) |
> | -------------- | ------------- | ------------- | ------------------------- | --------------------------- |
> | Qwen3-14B/1.7B | non-reasoning | 3.80±1.54     | 64.343                    | 48.883                      |
> |                | reasoning     | 4.17±1.39     | 72.249                    | 54.898                      |
> | Qwen3-14B/0.6B | non-reasoning | 3.68±1.50      | 70.703                    | 53.693                      |
> |                | reasoning     | 4.1±1.36      | 81.890                     | 62.157                      |
> | Qwen3-32B/1.7B | non-reasoning | 4.09±1.95     | 24.880                    | 39.430                      |
> |                | reasoning     | 4.62±.1.90     | 27.617                    | 43.373                      |

---

### Decision · Program_Chairs · 2025-09-17

**Decision:**

Accept (spotlight)

**Comment:**

Based on the comprehensive review process and author rebuttal addressing all major concerns, this meta-review recommends acceptance for this paper. All reviewers acknowledge the technical novelty of splitting speculative decoding workloads between edge and server GPUs, significantly reducing LLM serving costs while maintaining low latency. The proposed optimizations with proactive edge drafting and pipeline-aware scheduling demonstrate compelling improvements over server-only baselines. Reviewers initially raised valid concerns regarding cost analysis methodology, edge-device heterogeneity, fault tolerance, and minimum hardware requirements, all thoroughly addressed in the rebuttal with additional experiments and deployment strategy clarifications. The evaluation is rigorous, covering diverse models, tasks, and realistic WAN conditions. The work introduces a practical, cost-effective paradigm for LLM serving that leverages underutilized edge resources, advancing distributed systems research with tangible infrastructure benefits. Its alignment with NeurIPS's infrastructure focus and the clear consensus after rebuttal support acceptance.